# A Case of Isolated Cecal Necrosis Preoperatively Diagnosed with Perforation of Cecum

**DOI:** 10.3390/medicina55010009

**Published:** 2019-01-10

**Authors:** Atsushi Kohga, Kiyoshige Yajima, Takuya Okumura, Kimihiro Yamashita, Jun Isogaki, Kenji Suzuki, Akira Komiyama, Akihiro Kawabe

**Affiliations:** 1Division of Surgery Fujinomiya City General Hospital, Fujinomiya, Shizuoka Prefecture 418-0076, Japan; yaji@aurola.ocn.ne.jp (K.Y.); takuya330jp@yahoo.co.jp (T.O.); ymst13@yahoo.co.jp (K.Y.); jun_isogaki@hospital.fujinomiya.shizuoka.jp (J.I.); suzukik@hospital.fujinomiya.shizuoka.jp (K.S.); kaimai@hospital.fujinomiya.shizuoka.jp (A.K.); 2Division of Pathology Fujinomiya City General Hospital, Fujinomiya, Shizuoka Prefecture 418-0076, Japan; akomiyama-path@umin.ac.jp

**Keywords:** isolated cecal necrosis, laparoscopic-assisted surgery

## Abstract

Isolated cecal necrosis (ICN) is a rare condition which is developed under decreased mesenteric perfusion. Only a few dozen cases of ICN have been reported previously. The patient was a 59-year-old male with a previous history of atrial fibrillation. He presented to our emergency room with the chief complaint of lower abdominal pain. Computed tomography imaging revealed a dilated cecum and presence of free air. With a preoperative diagnosis of perforation of the cecum; an urgent surgery was conducted. Intraoperative findings revealed an ischemic change of the cecum and a laparoscopic-assisted ileocecal resection was performed. The pathological findings showed transmural ischemic change on the anti-mesenteric side of the cecum, and the diagnosis of ICN was achieved. Preoperative diagnosis of ICN is difficult because of its non-specific radiological features. In patients with right lower abdominal pain, ICN should be considered as a differential diagnosis especially if the patient has a comorbidity causing hypotension attack.

## 1. Introduction

Isolated cecal necrosis (ICN) is an infrequent variant of ischemic colitis [1]. Poor mesenteric perfusion, due either to systemic hypotension or to specific pharmacologic agents, is considered to play a role in the development of ICN [1,2,3]. According to previous reports, most patients who develop ICN have histories of congestive heart failure, cardiac surgeries or hemodialysis [1,2,3,4,5,6,7,8,9,10,11,12,13]. Preoperative diagnosis is difficult because of non-specific radiologic findings [2,14]. However, urgent surgical resection of the damaged intestine is required. Herein, we report a case of ICN who was preoperatively diagnosed with perforation of the cecum.

## 2. Case Report

The patient was a 59-year-old male who presented to our emergency department with a complaint of a 5-h duration of lower abdominal pain. He had a previous history of atrial fibrillation and had taken warfarin, digoxin, and bisoprolol. A physical examination revealed tenderness to palpation of the right lower abdomen without signs of peritoneal irritation. Laboratory data showed slight leukocytosis (WBC 8700/mm^3^) with slightly elevated C-reactive protein (CRP 0.43 mg/dL).

A computed tomography (CT) image revealed a dilated cecum surrounded by free air, while the appendix was intact (Figure 1). The preoperative diagnosis was perforation of the cecum. We planned to perform an urgent laparoscopic surgery.

During the laparoscopy, the cecum was found to be dilated and discoloured (Figure 2).

About 10 h after symptoms onset, we performed a laparoscopic-assisted ileocecal resection. After the resection of the damaged intestine, extracorporeal end to end anastomosis was performed. The specimen revealed localized ischemic change on the anti-mesenteric side of the cecum (Figure 3). Microscopically, the transluminal ischemic change was confirmed.

The lesion was diagnosed as isolated cecal necrosis. On day three postoperative, the patient developed a small intestinal perforation caused by sticking on the drainage tube, and he underwent ileostomy. After that, the patient progressed uneventfully, and the patient underwent closure of the ileostomy on the 51st postoperative day.

## 3. Discussion

One of the key factors for developing ICN is a presence of the comorbidity which causes decreased mesenteric perfusion [1,2,3,4,5]. ICN is divided into two groups according to the presence of comorbidities; Type I (spontaneously): The predicting factors responsible for the decreased mesenteric perfusion are not identifiable. Type II (secondary): Occur most often after an episode of systemic hypotension or decreased cardiac output or after aortic surgery [1]. Our patient could be classified as Type I because he did not have an apparent episode of systemic hypotension. He had a previous history of atrial fibrillation, and he had taken digoxin and bisoprolol. Digoxin has been reported as one of the causative agents of ICN as it decreases mesenteric perfusion [3]. On the other hand, occlusive ischemia was not considered as the cause of his disease, because he had taken warfarin.

Another key factor in developing ICN is the presence of a variation in cecum blood supply. The cecum is mainly supplied by the anterior and posterior caecal arteries. These arteries often arise from the vascular arcade between the ileal branch and colic branch of the ileocolic artery, while, in the others, these arteries arise directly from the ileal or colic branch [2,14,15]. If this arcade is absent, the cecum blood supply is considered to be deficient [2]. In addition, the vasa recta supplying the cecum are longest because this segment of bowel has the widest diameter, which makes the cecum vulnerable to ischemia [2]. Hunter et al. suggested that patients who develop ICN may have a particular variation in the cecal blood supply [15]. In our case, we ensured that a vascular arcade of the patient was absent by reviewing the preoperative CT images.

Guitart et al. suggested that preoperative CT findings suggesting ICN are: (1) thickening of the cecal wall; (2) abrupt transition between the ischemic cecum mural thickening and the ascending colon and (3) pneumatosis of the cecal wall [14]. However, these signs are non-specific and not always recognized as in our case. In addition, these non-specific findings sometimes lead to a misdiagnosis of cecal neoplasm [1,16]. On the other hand, we advocate that preoperative CT is essential for a surgeon to avoid misdiagnosis of this rare condition as acute appendicitis. According to previous reports, most ICN was preoperatively diagnosed with acute appendicitis [1,3,7,8,9,10,11,12,13,15,17,18]. We suppose that this misleading preoperative diagnosis could be prevented if a preoperative CT had been performed. In our case, we performed a diagnostic laparoscopy followed by a laparoscopic-assisted ileocecal resection with a mini-laparotomy. Diagnostic laparoscopy is considered as a useful option to make a definitive diagnosis and to implement a surgical strategy which includes incision type [18].

Previous reports have suggested that some cases of ICN showed poor prognosis [10]. On the other hand, many patients with ICN showed an uneventful postoperative outcome. Early diagnosis and urgent resection of the damaged intestine are essential for surgeons to improve the postoperative outcomes of ICN.

## 4. Ethics Statement

This case report is for academic communication only and is not for other purposes. This paper does not disclose the personal information of the patient and informed consent was obtained. The study protocol was approved by the institutional review board. The study does not include any animal experiments.

## 5. Conclusions

We reported a case of ICN. Preoperative diagnosis of ICN is difficult because of non-specific radiological features. In the patients with right lower abdominal pain, ICN should be considered as a differential diagnosis especially if the patient has comorbidity causing a hypotension attack.

## Figures and Tables

**Figure 1 medicina-55-00009-f001:**
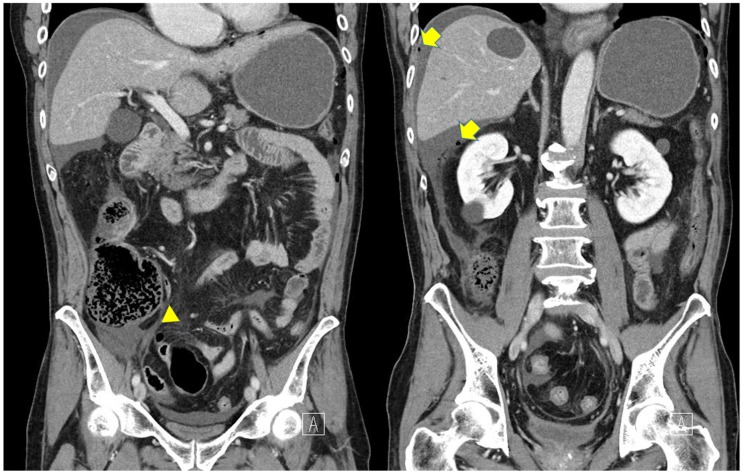
Computed tomography (CT) images revealed dilated cecum and free air. Arrowheads indicate appendix and Arrows indicate free air, respectively.

**Figure 2 medicina-55-00009-f002:**
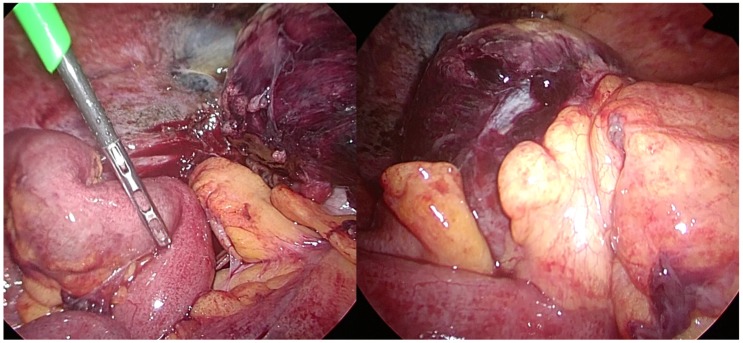
Laparoscopically, the cecum was found to be dilated with ischemic change, while the ileum and appendix were intact.

**Figure 3 medicina-55-00009-f003:**
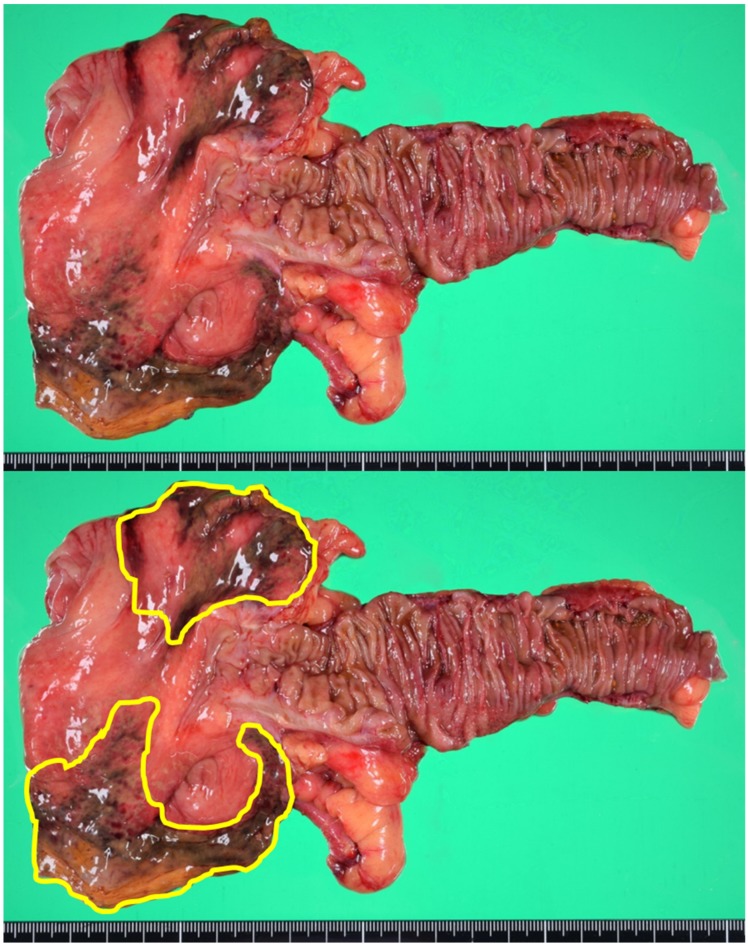
The specimen showed localized ischemic change on the anti-mesenteric side of the cecum. The enclosed area in the lower picture showing the damaged wall.

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
