# Peer review of "A Case of Isolated Cecal Necrosis Preoperatively Diagnosed with Perforation of Cecum"

_medicina, 2019, doi:10.3390/medicina55010009_

Round 1

Reviewer 1 Report

this is and interesting case reports. However all previous cases should be discussed and the differences have to be identified.

Author Response

Reply to reviewers

Dear Reviewer 1

We greatly appreciate your comments on our manuscript and recommendations regarding the references.

Reviewer: 1

Comments to the Author

this is and interesting case reports. However all previous cases should be discussed and the differences have to be identified.

Authors’ reply

Thank you for important suggestion. We had reviewed previously reported cases as much as possible and we had discussed about the differences or specificity of our case in Discussion section. Please let me know more detail, what kind of discussion we should do additionally. 

Reviewer 2 Report

This is a nicely written case report. I enjoyed reading it. I was suprised the incidence od cecal necrosis is so rare. Perhaps it is more common but less reported. 

I suggest the authors to write more about the procedure. what anastomosis you did (side to side/end to side? extra or intracorporeal) etc. 

I am not sure I understand what (WBC 87×102/μL) means. Can you present the value in some more clear way (like 11000/mm3 etc.)

What was the time interval between onset of symptoms and sugery?

Author Response

Dear Reviewer 2

We greatly appreciate your comments on our manuscript and recommendations regarding the references.

Reviewer: 2

Comments to the Author

This is a nicely written case report. I enjoyed reading it. I was suprised the incidence od cecal necrosis is so rare. Perhaps it is more common but less reported.

I suggest the authors to write more about the procedure. what anastomosis you did (side to side/end to side? extra or intracorporeal) etc.

Authors’ reply

Thank you for important suggestion.

In accordance with the reviewer’s comments, we rewrote the following sentences in the Case Report (page 5 line 12):

After the resection of damaged intestine, extracorporeal end to end anastomosis was performed.

I am not sure I understand what (WBC 87×102/μL) means. Can you present the value in some more clear way (like 11000/mm3 etc.)

Authors’ reply

Thank you for important suggestion.

In accordance with the reviewer’s comments, we rewrote the following sentences in the Case Report (page 5 line 6):

Laboratory data showed slight leukocytosis (WBC 8700/mm3) with slightly elevated C-reactive protein (CRP 0.43 mg/dl).

What was the time interval between onset of symptoms and sugery?

Authors’ reply

Thank you for your suggestion.

Time between onset of symptom and surgery was about ten hours.

In accordance with the reviewer’s comments, we rewrote the following sentences in the Case Report (page 5 line 11):

About 10 hours after symptoms onset, we performed laparoscopic assisted ileocecal resection.

Reviewer 3 Report

This is a short, direct, and to the point case study on a rare condition isolated cecal necrosis.   The background of this condition is presented well and the authors place their work within the context of other publications.   This is very important as it will provide those of interest with the most comprehensive breadth of this condition.   The manuscript is written well, it is clear, the necessary data provided and it is conservatively written.   This conservative nature is very important and well considered advice presented.  

Author Response

Dear Reviewer 3

We greatly appreciate your comments on our manuscript and recommendations regarding the references.

Reviewer: 3

Comments to the Author

This is a short, direct, and to the point case study on a rare condition isolated cecal necrosis.   The background of this condition is presented well and the authors place their work within the context of other publications.   This is very important as it will provide those of interest with the most comprehensive breadth of this condition.   The manuscript is written well, it is clear, the necessary data provided and it is conservatively written.   This conservative nature is very important and well considered advice presented. 

Authors’ reply

Thank you for your kind appreciation of our manuscript. We hope that you found this report suitable for publication in medicina.

Round 2

Reviewer 1 Report

this is an interesting case report. Other similar cases should be identified and discussed in the discussion section